# Malnutrition-Related Health Outcomes in Older Adults with Hip Fractures: A Systematic Review and Meta-Analysis

**DOI:** 10.3390/nu16071069

**Published:** 2024-04-05

**Authors:** Manuela Chiavarini, Giorgia Maria Ricciotti, Anita Genga, Maria Ilaria Faggi, Alessia Rinaldi, Oriana Dunia Toscano, Marcello Mario D’Errico, Pamela Barbadoro

**Affiliations:** Department of Biomedical Sciences and Public Health, Section of Hygiene, Preventive Medicine and Public Health, Polytechnic University of the Marche Region, 60126 Ancona, Italy; m.chiavarini@staff.univpm.it (M.C.); g.m.ricciotti@pm.univpm.it (G.M.R.); m.i.faggi@pm.univpm.it (M.I.F.); alessia.rinaldi@pm.univpm.it (A.R.); o.d.toscano@pm.univpm.it (O.D.T.); m.m.derrico@staff.univpm.it (M.M.D.)

**Keywords:** hip fracture, malnutrition, elderly, older adults, mortality, mobility limitation

## Abstract

Hip fracture is a common condition in older adults, leading to disability and mortality. Several studies have demonstrated the association between nutritional status and the risk of a negative health outcome after fractures. In this systematic review, we evaluated the association between malnutrition and mortality, changes in mobility/living arrangements, and postoperative complications, such as delirium, in older patients with hip fractures. A literature search on the PubMed, Web of Science, and Scopus databases, up to September 2023, was conducted to identify all studies involving older subjects that reported an association between MNA/GNRI/PNI/CONUT and health outcome after hip fracture. Meta-analysis was performed by a random-effects model using risk values (RR, OR, and HR) extracted from the 14 eligible selected studies. Malnutrition significantly increased the risk of any analyzed adverse outcome by 70% at 1 month, and up to 250% at 1 year. Malnutrition significantly increased delirium risk by 275% (OR = 2.75; 95% CI 1.80–4.18; *p* ≤ 0.05), mortality risk by 342% (OR = 3.42; 95% CI 2.14–5.48; *p* ≤ 0.05), mortality hazard risk by 351% (HR = 3.51; 95% CI 1.63–7.55; *p* ≤ 0.05) at 1 month, and transfer-to-more-supported-living-arrangements risk by 218% (OR = 2.18; 95% CI 1.58–3.01; *p* ≤ 0.05), and declined mobility risk by 41% (OR = 1.41; 95% CI 1.14–1.75; *p* ≤ 0.05), mortality risk by 368% (OR = 3.68; 95% CI 3.00–4.52; *p* ≤ 0.05), and mortality hazard risk by 234% (HR = 2.34; 95% CI 1.91–2.87; *p* ≤ 0.05) at 1 year. Malnutrition of older patients increases the risk of death and worsens mobility and independence after hip fractures. The results of the present study highlight the importance of nutritional status evaluation of older subjects with hip fractures in order to prevent potential adverse outcomes (Registration No: CRD42023468751).

## 1. Introduction

Hip fractures in older people is a substantial concern to public health services and society [1,2].

In fact, this type of fracture, in the upper part of the femur, becomes more common with aging. This is because aging-related bone fragility is exacerbated by structural deterioration and bone loss. The combination of both skeletal fragility and a greater propensity to fall leads to a high occurrence of hip fractures with advancing age [1].

It was estimated that hip fracture affected 1.6 million persons globally in 2000, and by 2050, this number is expected to reach 6.3 million [3].

Hip fracture is associated with the development of negative consequences, such as disability, depression, cardiovascular diseases, and, consequently, mortality [2]. Mortality within 1 year ranges from 20% to 40% in older patients [4,5].

Finding the risk factors for functional loss following hip fracture may help treat postoperative problems more effectively and lower the expenses of helping patients who become dependent on assistance due to institutionalization or loss of autonomy [6].

A recent systematic review found a range from 4% to 39.4% of hip fracture patients who were malnourished when they were admitted to hospital [7].

After a fracture, older patients who are undernourished may only partially regain their previous degree of independence in carrying out daily tasks and they have a reduced capacity to return to their pre-fracture functional status [8].

Malnutrition has been identified as a patient’s poor nutritional status and as an important and modifiable prognostic factor for several medical conditions.

Due to its modifiability and the potential for early nutritional intervention to promote hip fracture recovery, malnutrition is a condition of great concern when combined with hip fracture [9].

Malnutrition can be detected by several validated tools (https://guidelines.espen.org/espen-web-app/home/, accessed on 26 September 2023). In particular, the following are the most used for older patients or surgical patients within the hospital setting: the Mini Nutritional Assessment Short Form (MNA-SF), the Mini Nutritional Assessment Long Form (MNA-LF), the Geriatric Nutritional Risk Index (GNRI), the Prognostic Nutritional Index (PNI), and the Controlling Nutritional Status (CONUT) score.

The MNA is a standardized and validated instrument to identify protein–energy malnutrition or the risk of malnutrition in older patients in various care settings and has already been proven as a useful diagnostic tool among older orthopedic patients [10,11].

The GNRI is a tool used for assessing the nutritional status of older individuals considering both body weight and serum albumin levels to evaluate the risk of malnutrition and associated complications in geriatric populations. The GNRI score ranges from 0 to 100, with higher scores indicating better nutritional status [12].

The PNI was developed to investigate the relationship between nutritional status and outcomes in surgical patients. It is based on serum albumin levels and total lymphocyte count. Lower PNI scores indicate poorer nutritional status, while higher scores suggest better nutritional status [13].

The CONUT score is used for nutritional assessment in clinical practice, in particular, for patients with various medical conditions. It considers three parameters: serum albumin levels, total lymphocyte count, and total cholesterol concentration. It was first proposed in 2005 as a means of evaluating the nutritional status and predicting outcomes in hospitalized patients [14].

Two-thirds of older patients are at a particular nutritional risk or are malnourished, with a wide impact on their overall health, physical functioning, and quality of life [15].

Poor nutritional status has been associated with several negative clinical outcomes, such as postoperative complications, pressure ulcers, functional dependence, walking impairment, impaired quality of life, and mortality [9,16].

A recent meta-analysis [17] examined association between nutritional indices and mortality after hip fracture, finding that patients with low GNRI or low MNA-SF scores had a significantly higher risk of mortality compared to those with higher scores, even though it was not assessed for different follow-up periods.

To the best of our knowledge, there are no systematic reviews and meta-analyses that investigate the association between malnutrition and health outcomes in consideration of different follow-up intervals from hip fracture; therefore, the aim of our systematic review is to evaluate the association between malnutrition and selected health outcomes in patients affected by hip fractures, in particular: mortality, mobility, changes in living arrangements, and postoperative complications.

## 2. Materials and Methods

The present meta-analysis was conducted following the MOOSE (Meta-analysis Of Observational Studies in Epidemiology) guidelines [18] and PRISMA statement [19].

The protocol of this study has been recorded in the International Prospective Register of Systematic Reviews (www.crd.york.ac.uk/PROSPERO/ (accessed on 13 October 2023), Registration No: CRD42023468751).

### 2.1. Search Strategy and Data Source

We carried out a systematic literature search up to 26 September 2023 through the PubMed (http://www.ncbi.nlm.nih.gov/pubmed/), Web of Science (http://wokinfo.com/), and Scopus (https://www.scopus.com/) databases to identify original articles on the association between malnutrition and selected health outcomes in older individuals with hip fractures.

The literature search included the following medical subject headings (MeSH) and keywords: (“Nutritional Status”[Mesh] OR MNA[Title/Abstract] OR CONUT[Title/Abstract] OR PNI[Title/Abstract] OR GNRI[Title/Abstract]) AND (risk) AND (outcome) AND (((hip[Title/Abstract] OR femoral[Title/Abstract]) AND (injury[Title/Abstract] OR fracture[Title/Abstract])) OR “Proximal Femoral Fractures”[Mesh] OR “Femoral Neck Fractures”[Mesh] OR “Hip Fractures”[Mesh]).

The different associations of keywords combined with Boolean operators used for each database are shown in Appendix A.

No publication date limitation was applied but due to translation restrictions, only English-language studies were eligible.

We manually examined the reference lists of selected articles and recent relevant reviews to identify possible additional relevant publications.

### 2.2. Eligibility Criteria

Only the following tools were considered for inclusion: MNA-LF, MNA-SF, GNRI, PNI, and CONUT score.

Articles were included if they met the following criteria: (i) evaluated the relationship between malnutrition, derived by indices such as MNA-LF, MNA-SF, GNRI, PNI, or CONUT score, and health outcome in older patients after hip fractures; (ii) used a case-control, prospective or cross-sectional study design; (iii) reported odds ratio (OR), relative risk (RR), or hazard ratio (HR) estimated with 95% confidence intervals (CIs).

For each potentially included study, two investigators independently carried out the selection, data extraction, and quality assessment. Disagreements were resolved by discussion or in consultation with a third author. Although useful to have background information, reviews and meta-analyses were excluded. No studies were excluded for weakness of design or data quality.

### 2.3. Data Extraction and Quality Assessment

For each selected study, we extracted the following information: first author’s last name, year of publication, country, study design, sample size, population characteristics (sex, age), duration of follow-up for cohort studies, type of health outcome evaluated (mortality, mobility, changes in living arrangements, and postoperative complications), health outcome assessment method, type of nutritional assessment and categories of nutrition (at risk of malnutrition, malnourished), risk estimates with 95% CIs for the different categories of nutrition, *p*-value for trend, and confounding factors adjustment. When multiple estimates were reported in the article, we extracted those adjusted for the most confounding factors. The Newcastle–Ottawa Scale (NOS) was used to assess the quality of the literature for cohort and case-control studies using a 9-star system, as shown in Appendix A. The full score was 9 and a total score ≥7 was used to indicate a high-quality study [19].

### 2.4. Statistical Analysis

The estimated overall effect-size statistic was the average of the logarithm of the observed OR associated with the risk of malnutrition or malnourishing versus the normal state of nutrition. If the reference category was not normal, we reported the results in this form.

Concerning outcome, if it was presented in a positive form such as “survival” or “free walking ability”, the results were inverted or obtained from data extracted from the original text to obtain a negative outcome (“mortality” or “decreased walking ability”).

Change in living arrangements refers to an increase in level of care: living independently at home, living at home with organized home care, living in assisted living accommodation, and living in an institution.

The analysis was performed using the random-effects model to calculate the summary OR and 95% CIs. If the study reported mortality outcome as HR, OR with 95% CI was calculated from data extracted from the original text. The overall analysis of Any Health Outcome was conducted using OR, but for mortality, it was performed using both OR and HR separately.

We restricted this analysis to the following nutritional indexes: MNA-LF, MNA-SF, GNRI, PNI, and CONUT score, and to the following health outcomes: mobility, mortality, living arrangements, and postoperative complications, both overall and for different follow-up periods.

Stratification analysis for at risk of malnutrition and malnutrition groups was performed for observations with MNA and MNA-SF nutritional indexes, while those with GNRI, PNI, and CONUT score were included in the overall analysis.

An additional stratification was carried out for the study design (case-control or cohort study) and for age over 75 years.

The chi-square-based Cochran’s Q statistic and the I^2^ statistic were used to evaluate heterogeneity in results across studies [20].

The I^2^ statistic yields results ranged from 0% to 100% (I^2^ = 0–25%, no heterogeneity; I^2^ = 25–50%, moderate heterogeneity; I^2^ = 50–75%, large heterogeneity; and I^2^ = 75–100%, extreme heterogeneity) [21]. The results of the meta-analysis may be biased if the probability of publication is dependent on the study results.

We used the methods of Begg and Mazumdar [22] and Egger et al. [23] to detect publication bias. Both methods tested for funnel plot asymmetry, with the former being based on the rank correlation between the effect estimates and their sampling variances and the latter on a linear regression of a standard normal deviate on its precision.

If a potential bias was detected, we further conducted a sensitivity analysis to assess the strength of combined effect estimates, the possible influence of the bias, and to have the bias corrected. We also conducted a sensitivity analysis to investigate the influence of a single study on the overall risk estimate, by omitting one study in each turn. We considered the funnel plot to be asymmetrical if the intercept of Egger’s regression line deviated from zero, with a *p*-value < 0.05.

The analyses were performed using the ProMeta 3 statistical program and the calculations on data extracted from the original papers were performed using STATA 13.

## 3. Results

### 3.1. Study Selection

The literature search revealed 68 studies from the PubMed database, 47 from Web of Science, and 64 from Scopus.

After removing duplicates (n = 55), we identified 124 records screened for title and abstract revision (Figure 1). Among these, 76 articles were excluded (reviews, pooled or meta-analysis, commentary, and case studies).

Therefore, 48 studies were subjected to full-text revision.

We selected 5 additional items through the reference lists of both selected articles and recent relevant reviews [24,25,26,27,28,29,30,31,32,33,34,35,36,37,38,39,40,41,42,43,44,45,46,47,48,49,50,51,52,53,54,55,56,57,58,59,60,61,62,63,64,65,66,67,68,69,70,71,72,73,74,75,76].

Subsequently, 17 articles were also excluded because they did not meet the inclusion criteria as follows: 6 studies were not observational studies, 2 studies did not evaluate the impact on hip fracture, 3 studies did not consider the nutritional assessment, 3 studies did not investigate the association between nutritional assessment and outcomes, 2 studies did not report subjects aged ≥ 65 years old, and 1 study did not report the outcome.

At the end of the selection process, 36 studies were eligible for inclusion in the systematic review [25,28,30,31,33,34,36,38,39,40,41,43,44,45,46,48,49,50,51,53,54,56,58,59,60,61,64,67,68,69,70,71,72,73,75,76] and 14 studies were eligible for inclusion in the meta-analysis [34,36,39,43,49,51,53,56,64,67,68,69,70,72].

### 3.2. Study Characteristics and Quality Assessment

The general characteristics of the 14 studies included in the meta-analysis evaluating the association between malnutrition and health outcome in older adults with hip fractures and are shown in Table 1. Studies were conducted in Finland [34,43,49], the Netherlands [36], Italy [39], China [51,64,68,70], Taiwan [53], United States [56], Japan [67], Austria [69], and Spain [72]. There were 11 cohort studies [34,36,39,43,49,51,56,64,68,70,72] and 3 were case-control studies [53,67,69]. These studies were published between 2015 and 2023. All of these 14 studies were conducted in people aged ≥ 65 years old: women were more involved than men; regarding age, 5 studies [36,56,67,68,69] examined patients > 75 years old.

The nutritional status of all the participants was evaluated by different nutritional indexes: in 6 studies [34,39,43,49,69,72] by MNA-SF, 4 studies [53,56,67,68] used GNRI, 3 studies [51,64,68] used PNI, 2 studies [36,43] used MNA-LF, and 1 study [70] used CONUT score.

The health outcomes investigated were mortality in 11 studies [34,36,43,49,53,56,64,67,68,69,72], mobility in 6 studies [34,43,49,56,68,70], living arrangements in 2 studies [34,43], and complications (postoperative delirium) in 2 studies [39,51].

The study design was prospective in 10 studies [34,39,43,49,51,56,64,68,70,72] and retrospective in 4 studies [36,53,67,69].

Mortality assessment was obtained through electronic medical records [34,36,43,53,67,69,72] or telephone interview [49,56,64].

Changes in living arrangements or in mobility level were collected by telephone interview [34,43,49,56], using the Functional Independence Measure-locomotion (FIM-L) Scale in one study [70]. Postoperative delirium was evaluated according to the criteria by the Confusion Assessment Method (CAM) [39,51].

The follow-up considered in this analysis was 1–45 months. In particular, the follow-up for mortality was at ≤1 month in 4 studies, ≤3 months in 6 studies, ≤4 months in 8 studies, ≤6 months in 8 studies, and ≤12 months in all studies, except 1 study [64]; for mobility, follow-up was at ≤1 month in 2 studies, ≤4 months in 5 studies, and ≤12 months in all studies; for living arrangements, follow-up was at ≤1 month in 1 study, ≤4 months in 2 studies, and ≤12 months in all studies; and for complications (postoperative delirium), follow-up was only at ≤1 month.

In one study [64], the follow-up period was up to 45 months, so it was included in the overall analysis of any health outcome and mortality, but not in the follow-up stratification. One article [67] reported OR for “lower GNRI”, with no reference group; therefore, it was included only in the overall analysis of any health outcome and mortality. In one paper [70], the reference group was normal/mild, so it was not included in the stratification analysis, but only in the overall analysis of any health outcome and mobility.

### 3.3. Meta-Analysis

A comprehensive meta-analysis was conducted to investigate the association between nutritional status and various health outcomes among older patients with hip fractures. The analysis included the comparison between malnutrition, at-risk status, and all other nutritional categories, while the health outcomes examined included mortality, living arrangements, mobility, and postoperative delirium, as shown in Table 2. The overall analysis revealed a significant association between being in one of the risk categories of the nutritional tools examined and adverse health outcomes (OR = 2.42; 95% CI 2.07–2.83; *p* ≤ 0.05).

The odds ratio showed an increasing trend with longer follow-up periods (OR = 1.70; 95% CI 1.36–2.13; *p* ≤ 0.05 at ≤1 month, OR = 1.86; 95% CI 1.50–2.32; *p* ≤ 0.05 at ≤3 months, OR = 2.29; 95% CI 1.90–2.75; *p* ≤ 0.05 at ≤4 months, OR = 2.40; 95% CI 1.99–2.91; *p* ≤ 0.05 at ≤6 months, OR = 2.50; 95% CI 2.11–2.97; *p* ≤ 0.05 at ≤1 year).

While the category malnourished consistently showed significance since the first month of follow-up (OR = 3.01; 95% CI 1.75–5.17; *p* ≤ 0.05), the at-risk category emerged as significant from the fourth month of follow-up (OR = 1.67; 95% CI 1.37–2.02; *p* ≤ 0.05) and did so progressively over time (OR = 1.71; 95% CI 1.41–2.09; *p* ≤ 0.05 at 6 months and OR = 1.80; 95% CI 1.50–2.16; *p* ≤ 0.05 at 1 year). The heterogeneity was high in both pooled data (I^2^ = 85.94%) than in different months of follow-up data (I^2^ = 79.25% at 1 month, I^2^ = 82.56% at 3 months, I^2^ = 84.04% at 4 months, I^2^ = 85.82% at 6 months, and I^2^ = 86.37% at 1 year).

Figure 2 shows the forest plots of the selected studies that examined the associations between nutritional status and outcome at 1 month, 4 months, and 1 year.

One month after hip fracture, malnutritional status significantly increased delirium risk by 275% (OR = 2.75; 95% CI 1.80–4.18; *p* ≤ 0.05), mortality risk by 342% (OR = 3.42; 95% CI 2.14–5.48; *p* ≤ 0.05), and mortality hazard risk by 351% (HR = 3.51; 95% CI 1.63–7.55; *p* ≤ 0.05).

Four months from hip fracture, malnutrition significantly increased the probability of transfer to a higher level of care by 224% (OR = 2.24; 95% CI 1.52–3.31; *p* ≤ 0.05) and declined mobility risk by 32% (OR = 1.32; 95% CI 1.03–1.70; *p* ≤ 0.05), mortality risk by 379% (OR = 3.79; 95% CI 2.93–4.90; *p* ≤ 0.05), and mortality hazard risk by 253% (OR = 2.53; 95% CI 1.90–3.37; *p* ≤ 0.05).

One year after hip fracture, malnutrition significantly increased the probability of transfer to more supported living arrangements by 218% (OR = 2.18; 95% CI 1.58–3.01; *p* ≤ 0.05) and declined mobility risk by 41% (OR = 1.41; 95% CI 1.14–1.75; *p* ≤ 0.05), mortality risk by 368% (OR = 3.68; 95% CI 3.00–4.52; *p* ≤ 0.05), and mortality hazard risk by 234% (HR = 2.34; 95% CI 1.91–2.87; *p* ≤ 0.05). The heterogeneity was rather high in both pooled data (I^2^ = 85.94%), mobility (I^2^ = 51.31%), living arrangements (I^2^ = 64.54%), and mortality (I^2^ = 82.85% and I^2^ = 54.20% for OR and HR, respectively), while it was not present in delirium (I^2^ = 00.00%) (Table 2).

Furthermore, stratification by age group revealed significant findings in both any health negative outcome and mortality throughout the intervals of follow-up, while stratification by study type revealed significant findings in both case-control and cohort studies throughout the intervals of follow-up. Despite variations in the number of studies included, with cohort studies outnumbering case-control studies, the significance of the association remained consistent across study designs.

Because of the small amount of data, no further stratification according to gender, nutritional assessment, and different types of management of hip fractures were possible.

### 3.4. Sensitivity Analysis and Publication Bias

Table 2 reports results of both heterogeneity and publication bias tests. Considering the pooled data, on the basis of funnel plot symmetry for all outcomes), evidence of publication bias was detected at 1 month, 3 months, 4 months, 6 months, and 1 year. Accordingly, the corresponding statistical publication bias evaluation resulted in the *p*-value being significant for Begg/Egger’s test.

When results were stratified according to the specific outcome, a significant publication bias was observed only in mortality HR (Overall: Egger 0.005, Begg 0.022; 1-month All: Egger 0.12, Begg 0.042; 4-month All: Egger 0.011, Begg 0.020; 12-month All: Egger 0.007, Begg 0.023) and living arrangements (1-month All: Egger 0.035; 4-month All: Egger 0.00, Begg 0.002; 12-month All: Egger 0.00, Begg 0.01). No publication bias was detected in complication and mobility outcomes and in subgroup population aged >75 years. In addition, when results were stratified according to the type of outcome, a significant publication bias was observed only in the cohort studies on mortality HR (1, 4, and 12 months) and mortality OR (1 month).

A sensitivity analysis excluding Helminen 2017 [43], which caused asymmetry of the funnel plot, produced a combined risk estimate of:-2.32 (95% CI 1.65–3.28; *p* ≤ 0.05) with I^2^ = 0.00%, *p* = 0.568, and *p* = 0.212 and *p* = 0.174 for publication bias by the Egger and Begg methods, respectively, for any health outcome after hip fracture (OR) at 1 month in cohort study;-1.87 (95% CI 1.40–2.51; *p* ≤ 0.05) with I^2^ = 18.63%, *p* = 0.282, and *p* = 0.224 and *p* = 0.083 for publication bias by the Egger and Begg methods, respectively, for any health outcome after hip fracture (OR) at 3 months in cohort study;-1.88 (95% CI 1.16–3.06; *p* ≤ 0.05) with I^2^ = 47.91%, *p* = 0.124, and *p* = 0.137 and *p* = 0.174 for publication bias by the Egger and Begg methods, respectively, for mortality HR after hip fracture at 4 months in cohort study;-1.60 (95% CI 1.20–2.12; *p* ≤ 0.05) with I^2^ = 28.45%, *p* = 0.222, and *p* = 0.06 and *p* = 0.188 for publication bias by the Egger and Begg methods, respectively, for mortality HR after hip fracture at 1 year in cohort study.

## 4. Discussion

To the best of our knowledge, this is the first systematic review and meta-analysis to summarize the relationships between malnutrition and various health outcomes in older adults with hip fractures at different follow-up intervals. We considered data regarding all health outcomes and mortality, mobility, living arrangements, and complications, and analyzed them both together and separately for different follow-up intervals. Comparable results were observed in an earlier systematic review regarding the correlation between malnutrition and various clinical outcomes among hip fracture patients, including mortality rates, functional status, increased need for assisted living arrangements, and mobility levels [7,8]. We found a statistically significant association between malnutrition and any-health-negative-outcome risk in older subjects with hip fractures for all follow-up periods, with an increasing risk for longer follow-up periods. Malnutrition significantly increased any negative health outcome risk by 70% at 1 month to 250% at 1 year. While the malnourished category consistently showed significance since the first month of follow-up (OR = 3.01; 95% CI 1.75–5.17; *p* ≤ 0.05), the at-risk category emerged as significant from the fourth month of follow-up and did so progressively over time, suggesting a potential window for nutritional interventions to mitigate adverse outcomes following hip fracture. These findings underscore the critical role of nutritional status in influencing health outcomes among older individuals with hip fractures and suggest the importance of targeted interventions to improve nutritional status. With widespread rising life expectancy, there will be even more public health concern and an increasing need for initiatives in order to reduce this alarm.

The integration of both rehabilitation and nutritional therapy in older patients with hip fractures is a key strategy, resulting in decreased mortality and fewer postoperative complications, while also improving grip strength [77].

Therefore, it is advisable to endorse enhanced nutritional support alongside rehabilitation for individuals aged 65 years and older who have experienced hip fractures, aiming to mitigate mortality rates, minimize complications, and enhance ADL functionality [78].

We primarily focused on the association between hip fracture and malnutrition, overlooking the primary causes of hip fractures in older people: the role of diet and of osteoporosis. In fact, it is noted that diet is linked to the risk of osteoporotic fractures and plays a role in osteoporotic fracture healing and retaining their independence during osteoporosis prophylaxis [8,79,80].

### 4.1. Stratification Based on the Type of Outcome

Examining the mortality outcome, we observe a decline in the hazard ratio as the follow-up duration extends, with significance in the risk emerging noticeably from the fourth month onward, while the odds ratio concerning mortality rises with prolonged follow-up periods. The significance of living arrangements as an outcome becomes apparent from the fourth month, although it is important to underline the limited observations in the first month and the likelihood of patients still being hospitalized during this period. Similarly, the risk of declined mobility is significant from the fourth month, taking into account the probability of patients remaining hospitalized within the initial month. Lastly, postoperative delirium risk is significant in the only follow-up period available.

#### 4.1.1. Mortality

Our results are consistent with those of two prior meta-analyses, which found that being at risk of malnutrition and malnourished nutritional status according to MNA were both significantly associated with higher total mortality [17,81], and that patients with low GNRI or low MNA-SF scores had a significantly higher risk of mortality compared to those with higher scores. Other studies included in the systematic review found a significant correlation between nutritional status and mortality from a qualitative point of view: at risk of malnutrition/malnourished status is negatively correlated with 3, 6, and 12 months mortality according to MNA [48], and with 6 months mortality according to CONUT score [50] and GNRI [59,71].

#### 4.1.2. Postoperative Complications

A recent meta-analysis investigating the association between preoperative PNI and the incidence of POD in adult patients receiving surgery under general anesthesia demonstrated that a lower PNI was correlated with a higher risk of POD [82].

Other studies included in our systematic review evaluating postoperative complications such as urinary tract infection, heart failure, surgical site infection, refracture, pneumonia, arrhythmia, enteritis, liver dysfunction, implant failure, acute myocardial infarction, venous thromboembolism, confusion, delirium and dementia, decubitus, and the risk of falls found a significant correlation with GNRI and CONUT score [54,59,73].

#### 4.1.3. Mobility

A previous systematic review highlighted that interventions aimed at enhancing mobility post-hip fracture may result in clinically significant improvements in mobility and walking speed, both during hospitalization and in post-hospital settings, compared to standard care practices [83].

Studies included in the systematic review evidenced how decreased mobility level at discharge is significantly associated with MNA-SF, GNRI [45], and PNI [76].

Walking speed (m/sec) 14 days postoperative or at discharge is significantly lower in malnourished (MNA-SF) and major-risk and mild-risk patients (GNRI).

Also, rehabilitation effectiveness calculated using the FIM Scale is significantly lower in malnourished patients according to MNA-SF and GNRI [45,46], and they were more likely to be unable to walk 6 months postoperatively [58,59].

#### 4.1.4. Living Arrangements

Two studies included in the systematic review discovered that those who were malnourished according to MNA [33] and GNRI [58] exhibited notably lower scores in activities of daily living (ADL) within 48 h post-operation, upon discharge, and even at the 6-month mark post-discharge, but no significant relationship was found with discharge level of care [46].

### 4.2. Strengths and Limitations

The strength of this study lies in its approach of including several follow-up intervals for each outcome considered, which has not been seen before in a meta-analysis investigating health outcomes following hip fracture, as far as we are aware. Additionally, all included studies were of a high quality, scoring above 7 on the Newcastle–Ottawa Scale (NOS), ensuring reliability, even if we are aware of their possible limitations, especially in inter-rater agreement between observers of the NOS Scale [84,85,86].

Furthermore, this study stands out for evaluating various nutritional assessment tools to disclose associations with the outcomes examined.

Moreover, all articles included in this analysis were published within the last decade, reflecting the most current research in this field. However, several limitations should be acknowledged. The considerable heterogeneity observed across studies can be attributed to the diverse range of observations, nutritional assessments, and outcomes evaluated. Additionally, due to limited data availability, further stratification according to gender, nutritional assessment methods, or different types of hip fracture management was not possible.

Furthermore, while improvements in pre-surgery hospital stay and complications management have been noted, functional deterioration among hip fracture patients remains a concern in the literature. Specifically, although early surgery within 24 h or 48 h of admission has been associated with fewer postoperative complications, functional decline persists in many cases [87,88].

## 5. Conclusions

Older adult hip fracture patients exhibit a significant prevalence of malnutrition.

The present study underscores that malnutrition among older patients with hip fractures leads to an increased risk of mortality, impaired mobility, and loss of independence. Therefore, it is imperative to assess the nutritional status of older individuals with hip fractures and take proactive measures. Detecting malnutrition can aid in evaluating risks, initiating discussions about care goals, and making treatment decisions.

Our findings highlight the correlation between malnutrition and prognosis in older adults undergoing hip fracture surgery, supporting the utilization of nutritional assessment tools to furnish valuable prognostic insights. Incorporating these indicators into routine hospital admission procedures could prove beneficial in the management of hip fracture patients.

It is important to underline that all older patients can benefit from nutritional assessment and eventual appropriate interventions.

Nutritional rehabilitation stands out as a crucial strategy in preventing malnutrition, and the incorporation of nutritional substitution therapy presents a cost-effective and practicable solution. While there may be associated expenses with intensified nutritional interventions, the potential benefits are notable, including a potential reduction in the necessity for long-term care, thus alleviating the social burden.

Further research into nutritional interventions for older individuals with hip fractures is recommended and targeted interventions aimed at enhancing nutritional status during hip fracture management can be implemented based on these assessments.

## Figures and Tables

**Figure 1 nutrients-16-01069-f001:**
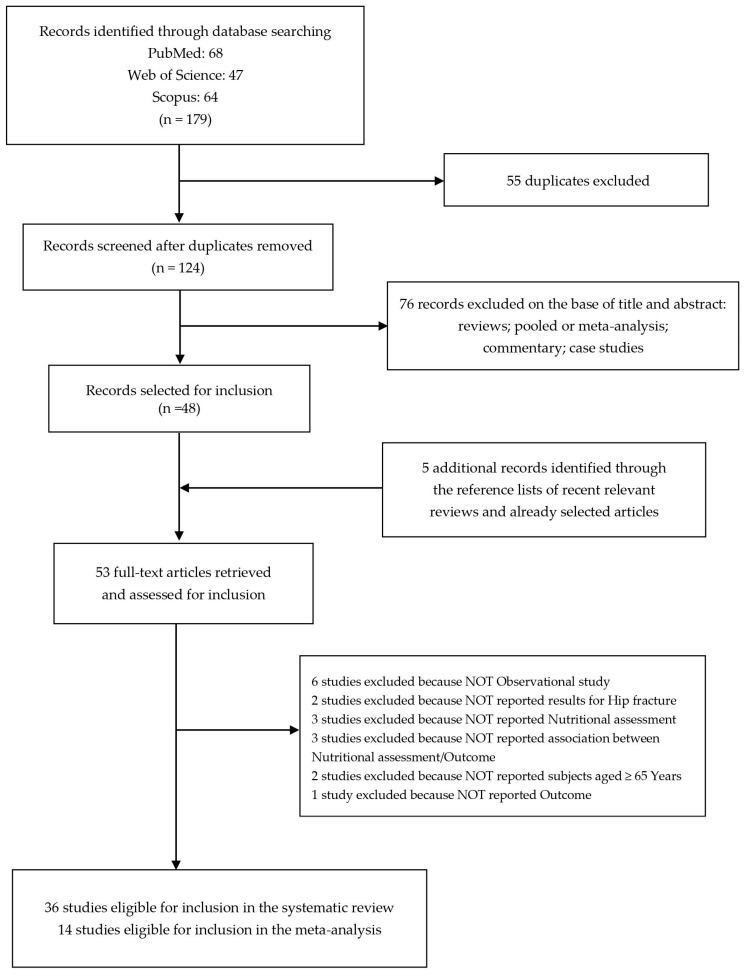
Flow diagram of the systematic literature search on malnutrition-related health outcomes in older adults with hip fractures.

**Figure 2 nutrients-16-01069-f002:**
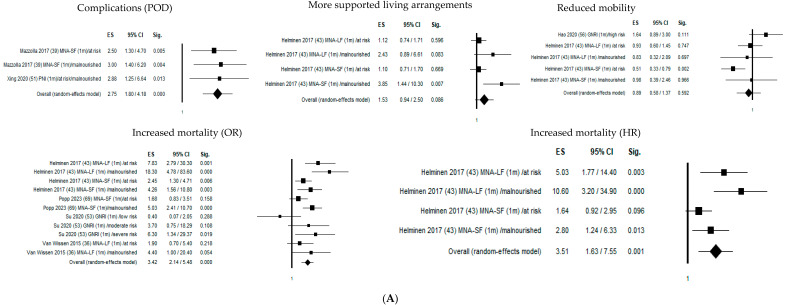
Forest plots of the selected studies that examined the associations between nutritional status and outcome at 1 month, 4 months, and 1 year. (**A**) Follow-up 1 month. (**B**) Follow-up 4 months. (**C**) Follow-up 12 months.

**Table 1 nutrients-16-01069-t001:** General characteristics of included studies in the meta-analysis of malnutrition and health outcome.

Author Year Country [Ref]	Study Design	Population Sample Size Period Age (y) Sex (M%)	Pre-Fracture Conditions: Comorbidities, Living, Mobility	NA MNA-LF MNA-SF GNRI PNI CONUT	Fracture Management	OUTCOMES (Follow-up) Mobility Living Arrangements Mortality Complications	OR/RR/HR (95% IC)	Matched or Adjusted Variables	NOS	Calculated OR (95% IC)—Unadjusted
Nuotio 2015 Finland [34]	Prospective	472 patients Period: January 2010–December 2012 Age > 65 M: 24.8%	Patients unable to move 2.3%, living in an institution 14%	MNA-SF At baseline: At risk 42% Malnourished 9%	Hip fracture Surgery	Mobility 4 m Living arrangements 4 m Mortality 4 m	Ref = Normal At risk OR = 2.03 (1.24–3.31) Malnourished OR = 2.32 (0.91–5.89) At risk OR = 2.42 (1.25–4.66) Malnourished OR = 6.10 (2.01–18.5) At risk HR = 1.32 (0.77–2.26) Malnourished HR = 2.16 (1.07–4.34)	Age, sex, ASA grade, pre-fracture memory disorder, and pre-fracture living arrangements	8	At risk OR = 3.18 (1.81–5.68) Malnourished OR = 5.22 (2.22–11.90)
Van Wissen 2015 The Netherlands [36]	Retrospective	226 patients Period: March 2008–July 2010 Age > 75 M: 26.9%	-	MNA-LF At baseline: At risk 27% Malnourished 5%	Hip fracture Surgery	Mortality in hospital 12 m	Ref = Normal At risk OR = 1.9 (0.7–5.4) Malnourished OR = 4.4 (1.0–20.4) At risk HR = 1.6 (0.9–3.0) Malnourished HR = 2.7 (1.1–7.0)	Age and sex	8	At risk OR = 1.14 (0.51–2.50) Malnourished OR = 2.85 (0.63–12.10)
Mazzola 2017 Italy [39]	Prospective	415 patients Period: September 2012–April 2016 Age > 70 (mean 84.0 ± 6.6) M: 25.5%	History of dementia 17.8%, psychotropic drugs 57.3%	MNA-SF At baseline: At risk 44.6% Malnourished 18.8%	Hip fracture Surgery	Complications (Postoperative delirium, 7 days)	Ref = Normal At risk OR = 2.5 (1.3–4.7) Malnourished OR = 3.0 (1.4–6.2)	Age, sex, CCI, ADL, MMSE score, history of dementia, psychotropic drug use, and ASA grade	9	
Helminen 2017 Finland [43]	Prospective	594 patients Period: December 2011–November 2014 Mean age: 84 (65–100) M: 28.5%	Pre-fracture diagnosis of memory disorder 32%, independent mobility 51%, living in own home 72%, ASA grade > 3 84.5%	MNA-SF At baseline: At risk 40% Malnourished 7% ___________ MNA-LF At baseline: At risk 58% Malnourished 7%	Hip fracture Surgery	Mobility 1 m 4 m 12 m Living arrangements 1 m 4 m 12 m Mortality 1 m 4 m 12 m _________________ Mobility 1 m 4 m 12 m Living arrangements 1 m 4 m 12 m Mortality 1 m 4 m 12 m	Ref = Normal At risk OR = 0.51 (0.33–0.79) Malnourished OR = 0.98 (0.39–2.46) At risk OR = 1.31 (0.87–1.98) Malnourished OR = 1.64 (0.68–3.95) At risk OR = 1.44 (0.91–2.29) Malnourished OR = 1.99 (0.70–5.67) At risk OR = 1.10 (0.71–1.70) Malnourished OR = 3.85 (1.44–10.3) At risk OR = 1.59 (0.91–2.77) Malnourished OR = 8.20 (2.70–24.9) At risk OR = 1.38 (0.79–2.43) Malnourished OR = 7.70 (2.17–27.3) At risk HR = 1.64 (0.92–2.95) Malnourished HR = 2.80 (1.24–6.33) At risk HR = 1.90 (1.26–2.87) Malnourished HR = 2.76 (1.51–5.05) At risk HR = 1.88 (1.32–2.69) Malnourished HR = 2.95 (1.75–4.98) ____________________ At risk OR = 0.93 (0.6–1.45) Malnourished OR = 0.83 (0.32–2.09) At risk OR = 1.84 (1.21–2.79) Malnourished OR = 2.40 (0.94–6.12) At risk OR = 1.88 (1.18–2.99) Malnourished OR = 3.28 (0.97–11.0) At risk OR = 1.12 (0.74–1.71) Malnourished OR = 2.43 (0.89–6.61) At risk OR = 1.67 (0.96–2.90) Malnourished OR = 4.77 (1.51–15.1) At risk OR = 1.25 (0.73–2.14) Malnourished OR = 4.19 (1.05–16.6) At risk HR = 5.03 (1.77–14.4) Malnourished HR = 10.6 (3.20–34.9) At risk HR = 2.92 (1.64–5.19) Malnourished HR = 4.69 (2.23–9.86) At risk HR = 2.73 (1.70–4.40) Malnourished HR = 5.11 (2.75–9.50)	Age, sex, ASA grade, and fracture type	8	At risk OR = 2.45 (1.30–4.71) Malnourished OR = 4.26 (1.56–10.8) At risk OR = 2.93 (1.84–4.70) Malnourished OR = 4.64 (2.11–9.92) At risk OR = 2.77 (1.82–4.23) Malnourished OR = 5.20 (2.49–10.8) At risk OR = 7.83 (2.79–30.3) Malnourished OR = 18.3 (4.78–83.6) At risk OR = 4.88 (2.65–9.54) Malnourished OR = 9.08 (3.54–23.0) At risk OR = 4.27 (2.54–7.43) Malnourished OR = 11.06 (4.69–26.0)
Helminen 2019 Finland [49]	Prospective	265 patients Period: November 2015–March 2017 Mean age: 84 (65–103) M: 33.3%	Diagnosis of memory disorder 32%, independent mobility 91%, living in own home 73%, ASA grade >3 83.8%	MNA-SF At baseline: At risk 40% Malnourished 7%	Hip fracture Surgery	Mobility 4 m Mortality 4 m	Ref = Normal At risk OR = 1.63 (0.86–3.07) At risk HR = 1.38 (0.68–2.82) Malnourished HR = 4.37 (1.77–10.8)	Age, sex, ASA, and fracture type Age, sex, ASA and fracture type, mobility level, and living arrangements at baseline	8	At risk OR = 2.42 (1.11–5.37) Malnourished OR = 11.16 (3.28–37.8)
Xing 2020 China [51]	Prospective	163 patients Period: March 2014–April 2017 Age ≥ 65 (mean 72) M: 43%	ASA grade 3–4 36.8%	PNI	Hip fracture Surgery	Complications (Postoperative delirium, 7 days)	Ref = PNI-high group PNI-low group OR = 2.88 (1.25–6.64)	Age, pre-operative MMSE score, duration of operation, type of anesthesia, Hb, albumin, and lymphocyte count	7	
Su 2020 Taiwan [53]	Retrospective	678 patients Period: January 2009–December 2019 Age range: 69–89 M: 34.2%	DM 34.5%, HTN 66%, CAD 13.4%, CVA 13%, ESRD 5.3%	GNRI At baseline: Low risk 18.1% Moderate risk 26.4% Severe risk 18.7%	Femoral fracture Any treatment	Mortality in hospital	Ref = Normal Low risk OR = 0.4 (0.07–2.05) Moderate risk OR = 3.7 (0.75–18.29) Severe risk OR = 6.3 (1.34–29.37)	Age, sex, pre-existing comorbidities, and injury severity	9	
Hao 2020 USA [56]	Prospective	290 patients Period: 2004–2009 Mean age: 82 ± 7 M: 27%	ASA grade mean: 2.9 ± 0.6	GNRI At baseline: Some risk 33% High risk 34%	Hip fracture Surgery	Mobility Free walking ability 1 m 2 m Mortality 2 m	Ref = High risk No risk OR = 1.57 (0.88–2.82) No risk OR = 1.02 (0.54–1.19) No risk OR = 0.68 (0.21–2.25)	Age and sex	8	Ref = No risk Decreased walking ability High risk OR = 1.64 (0.89–3.00) High risk OR = 1.08 (0.57–2.03) High risk OR = 1.47 (0.44–4.76)
Feng 2022 China [64]	Prospective	195 patients Period: January 2012–December 2018 Mean age: 78 (70–90) M: 21.2%	HTN 49.2%, DM 25.1%, CAD 18.5%, arrhythmia 21%, CVA 19%, DVT 6.7%, pulmonary disease 11.3%	PNI At baseline: At risk/ malnourished 13.3%	Hip fracture Surgery	Survival (Long-term postoperative: mean follow-up 1339 ± 610 days)	Ref = Normal At risk/malnourished HR = 0.269 (0.085–0.859)	-	7	Mortality At risk/malnourished HR = 3.72 (1.16–11.76) At risk/malnourished OR = 2.68 (0.98–6.94)
Fujimoto 2022 Japan [67]	Retrospective	108 patients Period: February–July 2007 Mean age: 84 (78–89) M: 21.3%	Pre-injury dementia 44.4%	GNRI At baseline: Mean: 92.8 ±8.62	Hip fracture Surgery	Survival 12 m	Lower GNRI OR = 0.80 (0.68–0.93)	-	7	Mortality Lower GNRI OR = 1.25 (1.08–1.45)
Liu 2022 China [68]	Prospective	546 patients Period: December 2017–May 2021 Mean age: 75.19 ± 10.23 M: 31.3%	CCI > 4 24.4%, HTN 51.1%, polytrauma 14.1%	GNRI At baseline: Low/ moderate/ severe risk 43.8% ___________ PNI At baseline: Moderate/ high risk 52.9%	Hip fracture Surgery	Mobility Free walking ability 3 m Mortality 12 m ________________ Mortality 12 m	Ref = Normal Low/moderate/severe risk OR = 0.602 (0.383–0.947) Low/moderate/severe risk HR = 1.467 (0.937–2.297) ___________________ Ref = Low risk Moderate/high risk HR = 1.295 (0.814–2.060)	Age, type of fracture, CCI, gout, HTN, and Hb	8	Decreased walking ability Low/moderate/severe risk OR = 1.48 (0.85–2.58) Low/moderate/ severe risk OR = 1.35 (0.69–2.62) __________________ Moderate/high risk OR = 1.45 (0.73–2.92)
Popp 2023 Austria [69]	Retrospective	1080 patients Period: January 2018–November 2019 Mean age: 81.1 M: 30.5%	-	MNA-SF At baseline: At risk 41.2% Malnourished 14.54%	Proximal femur fracture Surgery	Mortality 1 m 3 m 6 m 12 m	Ref = Normal At risk OR = 1.68 (*p* > 0.05) Malnourished OR = 5.03 (*p* < 0.01) At risk OR = 2.35 (*p* < 0.01) Malnourished OR = 7.28 (*p* < 0.01) At risk OR = 2.73 (*p* < 0.01) Malnourished OR = 7.44 (*p* < 0.01) At risk OR = 3.35 (*p* < 0.01) Malnourished OR = 7.77 (*p* < 0.01)	-	7	95% IC were obtained from data extracted from the original text
Cheng 2023 China [70]	Prospective	1958 patients Period: October 2014–April 2019 Mean age: 76 (69–83) M: 33%	ASA 3–4 47.9%, CCI ≥ 3 8.7%, HTN 53.9%, DM 24.5%, CVA 35.6%, heart disease 33.5%, kidney disease 5.6%, surgical history 31.5%	CONUT At baseline: Moderate/ severe malnutrition 51.3%	Hip fracture Surgery	Mobility 6 m	Ref = Normal/mild malnutrition Moderate/severe malnutrition RR = 1.42 (1.12–1.80)	Operation type, anesthesia type, surgical duration, and perioperative blood transfusion	8	OR = 1.36 (1.07–1.74)
Sánchez-Torralvo 2023 Spain [72]	Prospective	300 patients Period: September 2019–February 2021 Age > 65 (mean 82.9 ± 7.1) M: 20.7%	Previous fracture 11.3%, CCI mean: 5.67 ± 1.91	MNA-SF At baseline: At risk 42% Malnourished 37.3%	Hip fracture Any treatment	Mortality 3 m 6 m 12 m	Ref = Normal At risk/malnourished OR = 6.36 (0.79–51.06) At risk/malnourished OR = 5.71 (1.28 25.36) At risk/malnourished OR = 3.81 (1.25 11.57)	Age, sex, and CCI	9	

NA: Nutritional assessment, ASA: American Society of Anesthesiologists, CCI: Charlson Comorbidity Index, ADL: Activities of daily living, MMSE: Mini-Mental State Examination, Hb: Hemoglobin, DM: Diabetes mellitus, HTN: Hypertension, CAD: Coronary artery disease, CVA: Cerebrovascular accident, ESRD: End-stage renal disease, DVT: Deep vein thrombosis. MNA-LF: Mini Nutritional Assessment Long Form [36,43]. Normal 24–30. At risk of malnutrition 17–23.5. Malnourished < 17. MNA-SF: Mini Nutritional Assessment Short Form [34,39,43,49,69,72]. Normal 12–14. At risk of malnutrition 8–11. Malnourished 0–7. GNRI: Geriatric Nutrition Risk Index [53,68]. Normal > 98. Low risk 92–98. Moderate risk 82–91. Severe risk < 82. [56] No risk ≥ 98. Some risk 92–98. High risk < 92. PNI: Prognostic Nutritional Index. [51] PNI-high group ≥ 47.45. PNI-low group < 47.45. [64] Normal > 38. Malnourished ⩽ 38. [68] Low risk > 45. Moderate risk 40–45. High risk < 40. CONUT: Controlling Nutritional Status [70]. Normal 0–1. Mild malnutrition 2–4. Moderate malnutrition 5–8. Severe malnutrition 9–12.

**Table 2 nutrients-16-01069-t002:** Results of stratified analysis of the health outcome risk estimates, after hip fracture, according to nutritional status.

	Combined Risk Estimate ^a^			Test of Heterogeneity			Publication Bias	
	N. ^b^	Value (95% CI)	*p*	Q	I^2^%	*p*	*p* (Egger Test)	*p* (Begg Test)
Any health outcome after hip fracture (OR)								
ALL (ALL + art 64, 67, 70)	75	2.42 (2.07–2.83)	<0.01	526.17	85.94	<0.01	<0.01	0.087
ALL >75 y (ALL + art 67)	19	2.43 (1.71–3.44)	<0.01	143.84	87.44	<0.01	0.073	0.600
Period ≤ 1 month ALL At Risk Malnourished	23 9 9	1.70 (1.36–2.13) 1.12 (0.93–1.36) 3.01 (1.75–5.17)	<0.01 0.233 <0.01	106.02 28.14 21.97	79.25 71.57 63.59	<0.01 <0.01 0.005	<0.01 0.211 0.516	0.170 0.118 0.677
Cohort	16	1.49 (1.19–1.87)	<0.01	74.48	79.86	<0.01	0.005	0.013
Case-control	7	2.61 (1.44–4.75)	0.002	11.41	47.42	0.076	0.865	0.652
ALL >75 y	5	2.39 (1.45–3.93)	0.001	7.00	42.85	0.136	0.538	0.142
Period ≤ 3 months ALL At Risk Malnourished	29 10 10	1.86 (1.50–2.32) 1.21 (0.99–1.49) 3.39 (2.00–5.73)	<0.01 0.060 <0.01	160.51 32.52 29.74	82.56 76.01 69.74	<0.01 <0.01 <0.01	<0.01 0.106 0.627	0.268 0.152 0.788
Cohort	20	1.48 (1.20–1.81)	<0.01	79.13	75.99	<0.01	0.001	0.012
Case-control	9	3.03 (1.82–5.05)	<0.01	21.59	62.95	0.006	0.560	0.532
ALL >75 y	10	2.29 (1.50–3.51)	<0.01	31.42	71.35	<0.01	0.978	0.531
Period ≤ 4 months ALL At Risk Malnourished	50 21 20	2.29 (1.90–2.75) 1.67 (1.37–2.02) 3.90 (2.82–5.40)	<0.01 <0.01 <0.01	306.96 117.99 45.10	84.04 83.05 57.87	<0.01 <0.01 0.001	<0.01 <0.01 0.931	0.076 0.156 0.399
Cohort	41	2.15 (1.78–2.59)	<0.01	239.40	83.29	<0.01	<0.01	0.004
Case-control	9	3.03 (1.82–5.05)	<0.01	21.59	62.95	0.006	0.560	0.532
Period ≤ 6 months ALL At Risk Malnourished	53 22 21	2.40 (1.99–2.91) 1.71 (1.41–2.09) 4.08 (2.98–5.58)	<0.01 <0.01 <0.01	366.79 132.07 50.36	85.82 84.10 60.29	<0.01 <0.01 <0.01	<0.01 <0.01 0.456	0.131 0.195 0.763
Cohort	42	2.17 (1.80–2.62)	<0.01	243.48	83.16	<0.01	<0.01	0.005
Case-control	11	3.36 (2.20–5.14)	<0.01	31.39	68.14	0.001	0.487	0.392
ALL >75 y	12	2.62 (1.75–3.92)	<0.01	48.97	77.54	<0.01	0.626	0.681
Period ≤ 12 months ALL At Risk Malnourished	72 30 29	2.50 (2.11–2.97) 1.80 (1.50–2.16) 4.35 (3.38–5.62)	<0.01 <0.01 <0.01	520.76 200.53 63.71	86.37 85.54 56.05	<0.01 <0.01 <0.01	<0.01 <0.01 0.128	0.212 0.498 0.499
Cohort	57	2.28 (1.93–2.71)	<0.01	335.57	83.31	<0.01	<0.01	0.004
Case-control	15	3.34 (2.34–4.77)	<0.01	50.23	72.13	<0.01	0.290	0.400
ALL >75 y	18	2.55 (1.83–3.57)	<0.01	81.72	79.20	<0.01	0.197	0.472
Mobility (OR)—Cohort								
ALL (ALL + art 70)	19	1.40 (1.16–1.69)	<0.01	36.97	51.31	0.005	0.383	0.248
Period ≤ 1 month ALL At Risk Malnourished	5 2 2	0.89 (0.58–1.37) 0.69 (0.38–1.24) 0.90 (0.47–1.74)	0.592 0.213 0.762	9.99 3.60 0.06	59.95 72.22 <0.01	0.041 0.058 0.804	0.543 - -	1.00 - -
Period ≤ 4 months ALL At Risk Malnourished	14 6 5	1.32 (1.03–1.70) 1.24 (0.81–1.89) 1.49 (0.98–2.28)	0.029 0.331 0.062	31.86 25.54 4.19	59.19 80.42 4.60	0.003 <0.01 0.381	0.440 0.649 0.966	0.784 0.573 1.00
ALL >75 y	3	1.40 (0.99–1.97)	0.055	0.94	<0.01	0.624	0.652	0.602
Period ≤ 12 months ALL At Risk Malnourished	18 8 7	1.41 (1.14–1.75) 1.33 (0.95–1.85) 1.66 (1.15–2.39)	0.002 0.096 0.006	36.97 28.79 5.77	54.01 75.68 <0.01	0.003 <0.01 0.450	0.272 0.550 0.249	0.306 0.216 0.293
Mortality (OR)								
ALL (ALL + art 64, 67)	39	3.54 (2.74–4.57)	<0.01	221.61	82.85	<0.01	<0.01	0.570
ALL >75 y (ALL+ art 67)	16	2.72 (1.80–4.09)	<0.01	134.34	89.23	<0.01	0.053	0.418
Period ≤ 1 month ALL At Risk Malnourished	11 4 4	3.42 (2.14–5.48) 2.50 (1.45–4.31) 5.61 (3.31–9.50)	<0.01 0.001 <0.01	21.12 4.92 3.12	52.65 39.04 3.72	0.020 0.178 0.374	0.496 0.400 0.503	0.938 0.497 0.497
Cohort	4	5.29 (2.34–11.96)	<0.01	7.79	61.51	0.050	0.015	0.042
Case-control	7	2.61 (1.44–4.75)	0.002	11.41	47.42	0.076	0.865	0.652
ALL >75 y	4	2.77 (1.50–5.11)	0.001	5.17	42.23	0.158	0.861	0.497
Period ≤ 3 months ALL At Risk Malnourished	15 5 5	3.44 (2.33–5.07) 2.40 (1.66–3.48) 6.29 (4.30–9.20)	<0.01 <0.01 <0.01	32.31 4.92 3.58	56.67 18.76 <0.01	0.004 0.295 0.466	0.845 0.415 0.892	0.961 0.624 1.000
Cohort	6	4.30 (2.19–8.43)	<0.01	10.54	52.54	0.061	0.239	0.091
Case-control	9	3.03 (1.82–5.05)	<0.01	21.59	62.95	0.006	0.560	0.532
ALL >75 y	7	3.02 (1.82–5.01)	<0.01	16.65	63.97	0.001	0.567	0.881
Period ≤ 4 months ALL At Risk Malnourished	23 9 9	3.79 (2.93–4.90) 2.82 (2.23–3.57) 6.31 (4.71–8.44)	<0.01 <0.01 <0.01	43.54 9.16 5.80	49.47 12.70 <0.01	0.004 0.329 0.670	0.447 0.747 0.485	0.853 0.835 0.297
Cohort	14	4.17 (3.12–5.56)	<0.01	21.05	38.24	0.072	0.043	0.090
Case-control	9	3.03 (1.82–5.05)	<0.01	21.59	62.95	0.006	0.560	0.532
Period ≤ 6 months ALL At Risk Malnourished	26 10 10	3.90 (3.07–4.95) 2.80 (2.30–3.42) 6.57 (5.11–8.46)	<0.01 <0.01 <0.01	52.90 9.18 6.11	52.74 1.96 <0.01	0.001 0.421 0.729	0.530 0.717 0.793	0.947 0.788 0.421
Cohort	15	4.19 (3.17–5.52)	<0.01	21.30	34.28	0.094	0.035	0.102
Case-control	11	3.36 (2.20–5.14)	<0.01	31.39	68.14	0.001	0.487	0.392
ALL >75 y	9	3.38 (2.21–5.16)	<0.01	26.56	69.88	0.001	0.502	0.677
Period ≤ 12 months ALL At Risk Malnourished	37 14 14	3.68 (3.00–4.52) 2.85 (2.38–3.41) 6.76 (5.52–8.29)	<0.01 <0.01 <0.01	96.70 17.17 9.58	62.77 24.30 <0.01	<0.01 0.191 0.728	0.733 0.565 0.645	0.774 0.298 0.784
Cohort	22	3.86 (3.00–4.96)	<0.01	46.17	54.52	0.001	0.035	0.076
Case-control	15	3.34 (2.34–4.77)	<0.01	50.23	72.13	<0.01	0.290	0.400
ALL >75 y	15	2.94 (2.07–4.19)	<0.01	59.83	76.60	<0.01	0.190	0.458
Mortality (HR)								
ALL (ALL + art 64)	21	2.36 (1.94–2.89)	<0.01	43.67	54.20	0.002	0.005	0.022
Period ≤ 1 month ALL (Cohort) At Risk Malnourished	4 2 2	3.51 (1.63–7.55) 2.63 (0.89–7.78) 5.06 (1.38–18.49)	0.001 0.081 0.014	9.20 3.36 3.25	67.39 70.20 69.28	0.027 0.067 0.071	0.012 - -	0.042 - -
Period ≤ 4 months ALL (Cohort) At Risk Malnourished	12 6 6	2.53 (1.90–3.37) 1.92 (1.40–2.63) 3.46 (2.38–5.04)	<0.01 <0.01 <0.01	23.34 8.23 6.78	52.87 39.23 26.26	0.016 0.144 0.237	0.011 0.423 0.063	0.020 0.573 0.091
Period ≤ 12 months ALL At Risk Malnourished	20 9 9	2.34 (1.91–2.87) 1.96 (1.59–2.41) 3.45 (2.67–4.46)	<0.01 <0.01 <0.01	42.85 10.69 8.96	55.66 25.14 10.71	0.001 0.220 0.346	0.007 0.576 0.244	0.023 0.835 0.297
Cohort	18	2.39 (1.92–2.98)	<0.01	41.62	59.16	0.001	0.007	0.017
Case-control	2	1.87 (1.13–3.10)	0.015	0.86	<0.01	0.353	-	-
ALL >75 y	4	1.51 (1.15–1.98)	0.003	1.99	<0.01	0.575	0.059	0.174
Living Arrangements (OR) Cohort								
Period ≤ 1 month ALL At Risk Malnourished	4 2 2	1.53 (0.94–2.50) 1.11 (0.82–1.50) 3.07 (1.52–6.20)	0.086 0.499 0.002	7.22 <0.01 0.41	58.43 <0.01 <0.01	0.065 0.954 0.521	0.035 - -	0.497 - -
Period ≤ 4 months ALL At Risk Malnourished	10 5 5	2.24 (1.52–3.31) 1.41 (1.08–1.85) 4.50 (2.80–7.25)	<0.01 0.011 <0.01	27.77 5.48 2.97	67.59 27.06 <0.01	0.001 0.241 0.563	<0.01 0.001 0.171	0.002 0.142 0.327
Period ≤ 12 months ALL At Risk Malnourished	14 7 7	2.18 (1.58–3.01) 1.36 (1.12–1.66) 4.75 (3.11–7.26)	<0.01 0.002 <0.01	36.67 5.60 3.61	64.54 <0.01 <0.01	<0.01 0.47 0.73	<0.01 0.004 0.296	0.01 0.051 0.453
Complications (Delirium)(OR) Cohort								
Period ≤ 1 month ALL	3	2.75 (1.80–4.18)	<0.01	0.15	<0.01	0.928	0.408	0.602

^a^ Risk estimates were calculated using the random-effects model; ^b^ Number of data used to calculate the risk.

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
