# Peer review of "Malnutrition-Related Health Outcomes in Older Adults with Hip Fractures: A Systematic Review and Meta-Analysis"

_nutrients, 2024, doi:10.3390/nu16071069_

Round 1

Reviewer 1 Report

Comments and Suggestions for Authors

The manuscript entitled “Malnutrition-related health outcomes in elderly with hip fracture: A Systematic Review and Meta-Analysis” presents interesting issue but some problems must be corrected.

Major:

The main problem with the presented study is the fact that Authors declare that their systematic review and meta-analysis “was conducted following the MOOSE (Meta-analysis of Observational Studies in Epidemiology) guidelines [18] and PRISMA statement [19]”, but in fact the manuscript is not prepared according to the recommendations of PRISMA, which are very specific and should be rigorously followed. Authors should get familiar with PRISMA checklist (http://prisma-statement.org/prismastatement/Checklist.aspx) and they should correct their manuscript to be prepared according to the checklist. E.g., the Abstract should include: background, objectives, data sources, study eligibility criteria, participants, and interventions, study appraisal and synthesis methods, results, limitations, conclusions and implications of key findings, systematic review registration number, while a number of elements is not presented in the Abstract of the submitted manuscript. However, Authors should correct the whole study (not only the Abstract Section).

Introduction:

Authors should present the results of the most important meta-analyses conducted so far, in order to present adequately the current state of knowledge and novelty of the conducted study.

Materials and Methods:

Authors should present their methods in details with all necessary information provided.

Results:

The green table is extremely hard to follow and Authors did not provide here the essential information about the included studies which are necessary to interpret the data (e.g. inclusion and exclusion criteria, the place where the study was conducted, major conclusions from the study) – Authors should divide information into few smaller tables (based on the presented characteristics) and include additional information which should be provided.

Lines 319-347 are not explained

Figures – are hard to follow (with a small size of letters and not self-explanatory) – they should be corrected

Discussion:

Authors should explain why did they decide to choose Newcastle-Ottawa Scale, which is not the most recommended tool and indicate related consequences within the limitations of the study

Conclusions:

This sections should be simple and without elements of discussion (references should not be included into this section) – Authors should try to formulate brief statements which may be based on their study.

Reviewer 2 Report

Comments and Suggestions for Authors

The manuscript describes a literature review with a meta-analysis of malnutrition and outcomes in older adults with hip fractures.

Replace the term “elderly” in the title with the preferred term “older adults” to avoid ageism

There are too many tables in the text -some should be relegated to an appendix. Consider:

·         Place Tables 1 and 2 in the appendix

·         Combine the funnel plots in Figure 2 (individual living arrangement, mortality plots) into a concise display and relegate the other data to an appendix

Describe living arrangements -unclear if this is a combination of activity level and location of care.

Typo line 415 – “live arrangement”

Conclusion -discuss the association identified between malnutrition and poor outcome in older adults with hip fractures

a)       There is a high prevalence of malnutrition in older adult hip fracture patients

b)      The identification of malnutrition may assist in risk assessment as well as goals of care discussions and treatment decisions in older adult hip fracture patients

c)       Nutritional assessment and intervention is indicated for all older adults

d)      More studies of nutritional interventions in older adults with hip fractures are needed, as suggested by [84]

Round 2

Reviewer 2 Report

Comments and Suggestions for Authors

1.       The phrases elderly people, elderly individuals, geriatric patients continue to appear throughout the manuscript.  This should be changed to “older adults” in all locations.  See lines 10, 27, 38, 44, 399, 574, 665, 667

2.       Line 155: Recommend changing: Change in living arrangements refers to an increase in level of care: Living at home….

3.       Recommend changing the term “ living arrangements” to “ higher level of care” throughout the manuscript.

4.       Line 395: Use lowercase for any, mobility.

5.       Line 422: Recommend changing “ malnutrition condition” to “ nutritional status” in the legend

6.       Figure 2 requires clarification of all headings: Reduced mobility, increased mortality, higher level of care

7.       Line 497: Rephrase: Malnutrition increased the probability of transfer to a higher level of care…

8.       Line 503: … while it was not associated with delirium

9.       Line 555: use malnutrition instead of nutritional status

10.   Line 625: Change to: discharge level of care

11.   Line 699: In the sentence at “ management.”  And delete lines 670-674

Comments on the Quality of English Language

minor changes, suggestions provided in comments above

Author Response

Thank you very much for taking the time to review this manuscript once again.

Regarding the first suggestion, we replaced the terms containing "elderly" with “older adults” in all locations throughout the manuscript.

We corrected line 155, but we decided not to change the term "living arrangements" because it's how the outcome was defined in the original articles.

We also corrected lines 395, 422, 497, 555, 625 and Figure 2 headings.

Regarding line 503, we replaced the term "apparent" with "present" as we think it's a better fit to describe heterogeneity.

We didn't understand your suggestion about line 699, which is part of the reference list.

Lastly, we decided to move lines 670-674 above because we think it's an appropriate information for the conclusion paragraph.